# Epidemiology of Δ8THC-Related Carcinogenesis in USA: A Panel Regression and Causal Inferential Study

**DOI:** 10.3390/ijerph19137726

**Published:** 2022-06-23

**Authors:** Albert Stuart Reece, Gary Kenneth Hulse

**Affiliations:** 1Division of Psychiatry, University of Western Australia, Crawley, WA 6009, Australia; gary.hulse@uwa.edu.au; 2School of Medical and Health Sciences, Edith Cowan University, Joondalup, WA 6027, Australia

**Keywords:** cannabis, cannabinoid, Δ8THC, teratogenesis, oncogenesis, carcinogenesis, cancer, cancerogenesis

## Abstract

The use of Δ8THC is increasing at present across the USA in association with widespread cannabis legalization and the common notion that it is “legal weed”. As genotoxic actions have been described for many cannabinoids, we studied the cancer epidemiology of Δ8THC. Data on 34 cancer types was from the Centers for Disease Control Atlanta Georgia, substance abuse data from the Substance Abuse and Mental Health Services Administration, ethnicity and income data from the U.S. Census Bureau, and cannabinoid concentration data from the Drug Enforcement Agency, were combined and processed in R. Eight cancers (corpus uteri, liver, gastric cardia, breast and post-menopausal breast, anorectum, pancreas, and thyroid) were related to Δ8THC exposure on bivariate testing, and 18 (additionally, stomach, Hodgkins, and Non-Hodgkins lymphomas, ovary, cervix uteri, gall bladder, oropharynx, bladder, lung, esophagus, colorectal cancer, and all cancers (excluding non-melanoma skin cancer)) demonstrated positive average marginal effects on fully adjusted inverse probability weighted interactive panel regression. Many minimum E-Values (mEVs) were infinite. *p*-values rose from 8.04 × 10^−78^. Marginal effect calculations revealed that 18 Δ8THC-related cancers are predicted to lead to a further 8.58 cases/100,000 compared to 7.93 for alcoholism and −8.48 for tobacco. Results indicate that between 8 and 20/34 cancer types were associated with Δ8THC exposure, with very high effect sizes (mEVs) and marginal effects after adjustment exceeding tobacco and alcohol, fulfilling the epidemiological criteria of causality and suggesting a cannabinoid class effect. The inclusion of pediatric leukemias and testicular cancer herein demonstrates heritable malignant teratogenesis.

## 1. Introduction

Δ8THC use and exposure is increasingly dramatically in many parts of the USA driven partly by the rubric of being “legal weed” [1]. Δ8THC differs from Δ9THC chemically by having the double bond on the eighth carbon on the C-ring whereas in Δ9THC, the double bond is on the ninth [2]. Δ8THC differs from Δ9THC biologically as its effects are milder than those of Δ9THC [2] notwithstanding, for which recent warnings concerning adverse effects have recently been issued from both the Centers for Disease Control (CDC) Atlanta, Georgia and the Food and Drug Administration (FDA) [3,4]. Like Δ9THC, Δ8THC is a partial agonist at CB1 receptors (CB1R) [2,5]. Its effects in overdose seem to closely parallel those of Δ9THC including sedation, nausea, vomiting, dysphoria, and hallucinations [2,6].

Several cannabinoids also have a long history as known genotoxic compounds. This has been demonstrated in the laboratory with toxic effects on chromosomes [7,8,9,10,11,12], DNA strands [13], DNA nucleoside bases [13], the epigenome [14,15,16], and intermediary metabolism, which drive epigenomic processes [17,18,19,20,21], being well-described. The adult cancer most strongly implicated with cannabis exposure is testicular cancer [22,23,24,25], but several childhood cancers including acute myeloid leukemia, rhabdomyosarcoma, and neuroblastoma have also been described in association with cannabis exposure [26]. Further interest in this issue has recently been aroused with the description that breast cancer, the most common cancer in many countries, is linked with cannabis exposure, along with thyroid, liver, and pancreatic tumors and acute myeloid leukemia [27]. Moreover, cannabis exposure has also been linked with the most common childhood cancer, acute lymphoid leukemia [28], and cannabis has also been shown to be a primary driver of total childhood cancer [28]. Since childhood cancers generally arise as a result of inherited genotoxic damage [29,30], such findings necessarily raise the concern of mutagenic processes with intergenerational impacts.

Like other cannabinoids [31,32,33,34,35], Δ8THC has also been shown to inhibit DNA synthesis [5,36,37,38,39]. Indeed, in one relative potency study, Δ8THC was shown to inhibit DNA synthesis more strongly and for a longer duration than other cannabinoids including Δ9THC [38]. Whilst this suggests an anti-cancer action for Δ8THC derivatives [36,37,39,40], its effects in vivo may be to potentiate carcinogenesis by interfering with DNA, RNA, and key nucleoprotein metabolism and DNA maintenance pathways [37,38]. As it has been shown long ago that the genotoxic moiety of cannabinoids resides in their central olevitol nucleus core structure [41,42], it is entirely possible that the described genotoxicity of many cannabinoids [13,43] is actually a class effect pertaining broadly to numerous cannabinoid derivates.

As important as malignant outcomes are, recent reports indicate that the stakes are likely much higher, even than cancer. Cannabinoids have recently been linked with numerous severe congenital anomalies [27,44,45,46,47]. Moreover, recent studies have confirmed that epigenotoxicity causally mediates the aging processes [48]. Such findings greatly extend the present concerns relating to genotoxicity horizontally throughout the community via food chain contamination and vertically with transgenerational temporality.

As the issue of the potential relationship of Δ8THC to cancer presently represents a knowledge gap, we sought to investigate relevant epidemiological evidence in the USA using the most recent data available. Our hypothesis of a potential relationship between Δ8THC and cancer was formulated prior to commencement of the study.

## 2. Methods

Data. Age adjusted cancer incidence data were accessed from the Surveillance Epidemiology and End Results (SEER) database held by the National Cancer Institute and the Centers for Disease Control (CDC) Atlanta Georgia using the SEER*Stat software from CDC [49]. State-level drug use prevalence data were accessed from the Substance Abuse and Mental Health Services Administration (SAMHSA) Restricted Data Access Scheme (RDAS) of the annual National Survey of Drug Use and Health (NSDUH), a large nationally representative survey with a 74.1% response rate [50]. Substances of interest were cigarettes, alcohol abuse or dependency (such as alcohol use disorder, AUD), last month of cannabis use, last year of narcotic analgesic abuse, and last year of cocaine use. Median household income, ethnicity, and population data was from the U.S. Census Bureau using the R Package tidycensus [51]. The ethnicities considered were Caucasian American, African-American, Asian-American, Hispanic-American, American Indian/Alaskan Native American (AIAN), and Native Hawaiian/Pacific Islander American (NHPI). Mean cannabinoid concentration in federal seizures was taken from published reports [52,53,54].

Derived data. Quintiles of substance exposure were calculated by dividing up the range of substance exposures across the whole period for which data were available. Estimates of state-level cannabinoid exposure were obtained by multiplying the state-level of cannabis use by the concentration of the cannabinoids found in Federal seizures.

Statistics. Data were processed in R Studio (1.4.1717) based on R (4.1.1), both obtained from the comprehensive R Archive Network (CRAN). The analysis was performed in October 2021. Data were manipulated with dplyr and graphs were drawn using ggplot2, both from the tidyverse software suite [55]. Maps were drawn using the R package simple features (sf) [56]. Heatmaps were drawn in gplots [57]. All graphs and graphs are original. Relative risk ratios (RR), attributable fraction in the exposed (AFE), and population attributable risk (PAR) and their confidence intervals were calculated on the categorical data using the R package epiR [58]. Multivariable panel regression was performed using the R-package plm [59] and allows for data to be considered in its space–time context (using the “twoway” effect), the temporal lagging to be used, and E-Values to be calculated. All panel models were inverse probability weighted using the ipw R package [60]. In all interactive models, a three-way interaction was introduced between tobacco, alcohol use disorder, and Δ8THC exposure. E-values, or expected values, allow for a quantification of the extent to which an observed association might possibly be attributed to external or extraneous uncontrolled covariates [61,62]. E-Values were calculated using the EValue R package [63]. The E-Value has a confidence interval reflecting its upper and lower bound. Minimum E-Values in excess of 1.25 are said to indicate causality [64] and E-Values exceeding nine are considered to be very high [65]. Marginal overall effect sizes for panel models was estimated using the R package margins [66]. All tumor types were considered simultaneously using an iterative analytical approach in purrr (from tidyverse [55]) and the above R Packages together with broom [67].

Data availability. Data have been made publicly available through the Mendeley data repository and can be accessed at this URL doi:10.17632/nhprw35ppb.1.

Ethical approval. Ethical permission for this study was granted through the University of Western Australia Human Research Ethics Committee on 24 September 2021 with HREC Number 2019/RA/4/20/4724.

## 3. Results

As shown in Appendix A, 14,598 age-adjusted cancer incidence rates were downloaded from the CDC/NCI SEER database for the years 2009–2017. This time range represents the range for which both cancer and cannabinoid concentration data are available. The 34 cancer types of interest are listed in Appendix A. This table also provides data on state-level substance exposure rates together with estimates of state-level cannabinoid exposure and ethnicity and median household income, which were the other input covariates. Time trends of the mean cannabinoid content of USA seizures of cannabis at the Federal level are shown in Appendix A. Most cannabinoids are noted to be rising, with the notable exception of cannabidiol. A map showing the estimates of state-level Δ8THC exposure across USA over time is shown in Appendix A. Figure 1 illustrates a map-graph of the log Δ8THC state-level exposure estimates.

### 3.1. Continuous Bivariate Analysis

Appendix A shows the incidence of the 34 cancers of interest as a gradient function of tobacco exposure. The graph successfully identified many tumors that are known to be tobacco related including lung, larynx, colorectal, cervical, bladder, and all (excluding non-melanoma skin) cancers, etc. Similarly, Appendix A shows 34 cancer types as a function of alcohol consumption. In this graph, bladder, gastric, post-menopausal breast, and esophageal cancer are all cancers known to be associated with alcohol, which were correctly identified.

Appendix A illustrates the gradients of these cancers with state-level estimates of Δ9THC exposure. Gastric, post-menopausal breast, breast, bladder, liver corpus uteri, and thyroid cancers were all positively identified, some of which have recently been reported [27]. Figure 2 makes a similar plot for the state-level estimates of Δ8THC exposure. In this graph, the slopes of the associations with cancers of the following sites were all obviously positive: corpus uteri, gastric, breast overall and post-menopausal breast, liver, anorectum, pancreas, and thyroid cancers. A formal analysis of these trend lines is shown in Appendix A, which demonstrates that 16, 11, 8, and 8, cancers had elevated minimum E-Values (mEV) each for tobacco, alcohol, Δ9THC, and Δ8THC, respectively.

### 3.2. Categorical Bivariate Analysis

Appendix A illustrate the bivariate comparisons of the highest and lowest exposure quintiles for these four substances by tumor type. A formal analysis of this categorical data by cancer and by substance is shown in Table 1 and Appendix A, which demonstrate that for tobacco, alcohol, Δ9THCm and Δ8THC, 16, 15, 11, and 12 cancer types demonstrated elevated minimum E-Values, respectively.

### 3.3. Multivariable Panel Regression

Having identified that several substances were related to cancer incidence, the next question related to their relative importance. Panel regression allowed for these issues to be considered in their native space–time context. An inverse probability weighted comprehensive additive model including tobacco, alcohol, analgesics, cocaine, Δ8THC exposure, median household income, and all ethnicities was considered. A total of 77 significant terms with positive regression coefficients are shown in Appendix A. Appendix A selects the fifteen tumor types with which Δ8THC exposure was positively associated. One notes here that this list is headed by tumors of the ovary, oropharynx, gastric cardia, and post-menopausal breast cancer, which have *p*-values as low as 1.02 × 10^−69^ and minimum E-Values ascending from 2.64 × 10^14^. Appendix A is summarized in Appendix A by covariate. The number of cancers implicated, the negative sum of the *p*-value exponents, and the sum of the minimum E-Value exponents are shown. To assist in thee comparison, these three metrics are shown ordered in Appendix A. It is clear from this Figure that Δ8THC exposure features close to the top of the list in Panel A, second top of the list in Panel B, and in first place (cumulative minimum E-Value exponents) in Panel C.

As it was of interest to consider the interactive effects of the tobacco:alcohol:Δ8THC interaction, an inverse probability weighted interactive model including this three-way interaction was also considered. A total of 202 significant positive terms are listed in Appendix A. Importantly, the first 43 terms in this table all include Δ8THC. Remarkably, 36 minimum E-Values were listed as infinite and 80 were above 1000. Table 2 extracts from this list 49 terms including Δ8THC and notes that 36 mEVs were infinite and 48/49 (97.9%) mEVs were above 100,000,000. Thirty cancers were included on this list. Table 3 summarizes these results by covariate in tabular form and there are presented graphically in Figure 3. Once again, terms including Δ8THC appeared toward the right of these bar charts. In Panel C, terms incorporating Δ8THC occupied the top four positions on the graph for cumulative mEVs, with the leading position taken by the tobacco:Δ8THC interaction.

### 3.4. Temporally Lagged Panel Models

#### 3.4.1. Two Years Temporal Lag

The same exercise may be conducted with all independent variables lagged by two years. An inverse probability weighted panel model including the same three-way tobacco:alcohol:Δ8THC interaction was therefore considered. A total of 179 positive and significant terms from such models are shown in Appendix A. Again. 39 terms were noted to have infinite mEVs and Δ8THC was noted to be included in interactive terms in the first 40 positions. Appendix A extracts the terms including Δ8THC and it may be observed that 32 (78.0%) mEVs were infinite, and all exceeded 10^12^. Twenty-eight different tumor types were included on this list. These data are summarized by covariate in Appendix A and are presented graphically in Appendix A. In this figure, terms including Δ8THC were noted to occupy the mid-range of the number of tumors implicated and the significance levels, but the highest range for the cumulative mEVs (Panel C).

#### 3.4.2. Four Years Temporal Lag

As cancer is a disease that is believed to usually have a long incubation period, it is also of interest to consider models lagged to four years. For these purposes, the above interactive inverse probability (IPW) weighted model lagged to four years in all independent covariates was considered. Appendix A presents the 182 significant positive terms from this model. Table 4 extracts the 49 terms including Δ8THC, which relate to 25 tumor types. Remarkably, 48 terms (97.9%) had positive mEVs and the remaining one was 2.30 × 10^123^. These data are summarized in tabular and graphical formats in Appendix A and Figure 4. In Panel A, terms including Δ8THC were found in the mid-position of the graph. In Panel B (cumulative significance levels), terms including Δ8THC occurred at the right end extreme of the graph. In panel C, which lists the cumulative mEVs, all four positions at the top of the graph were occupied by terms including Δ8THC.

#### 3.4.3. Marginal Effects

Given that the independent variables each have their own scale and in interactive models it can be difficult to tell what the overall or “marginal” effect of the covariates might be, it is of interest to consider the marginal effect of these covariates in additive, interactive, and lagged models. The marginal effect was calculated in units of the standard deviation of the dependent variable, the cancer rate. Appendix A shows the average marginal effect by tumor and substance type for the additive model. Fifteen cancer types were noted to have positive average marginal effects (AME). Appendix A undertook the same exercise for the interactive, two and four lag models and noted that 18, 17, and 10 cancers were associated with positive marginal effects. These findings are presented graphically in the heatmaps of Figure 5 and Appendix A.

Table 5 collates the AME data for the substances, shows the mean and standard deviation of the various tumor incidences, and calculates the change in the case numbers for each tumor. The totals are noted near the foot of the table. The applicable figures for “All cancers” are also provided. It is noted that the numbers for breast cancer exceeded that for all cancers as the incidence of that tumor type was calculated on a single sex. This table shows that the total number of cancers attributable to Δ8THC was an extra 8.58/100,000 compared to 7.93 for AUD and −8.48 for tobacco (for this group of cancers).

## 4. Discussion

### 4.1. Main Results

The study findings are remarkable as both the number of tumors to which the population Δ8THC exposure was related as well as the extraordinarily elevated degree of association demonstrated by the very high mEVs. In a straight bivariate continuous analysis, Δ8THC was shown to be significantly associated with eight cancers (corpus uteri, liver, gastric cardia, post-menopausal breast cancer, anorectum, pancreas, breast, and thyroid, Appendix A) and a comparison of the highest and lowest quintiles of exposure showed that 12 cancers were significantly related to Δ8THC exposure on categorical analysis (additionally: melanoma, corpus uteri, anorectum, acute myeloid leukemia, pancreas, breast, testicular, and oropharynx, Table 1). In an additive IPW panel model adjusting for all substances, ethnicity, and median household income, 15 cancers were related to population Δ8THC exposure including stomach, Hodgkins, acute lymphoid leukemia, brain, breast, cervix, colorectal, and myeloma in addition to those above-mentioned (Appendix A). An interactive panel model found significant positive terms for 30 cancer types which, in addition to those listed above included all cancers (excluding non-melanoma skin cancers), lung, gall bladder, Non-Hodgkins lymphoma, kidney, chronic myeloid and lymphoid leukemias, vulva and vaginal, esophagus, and prostate cancers.

Consideration of the category “All cancers” (excluding non-melanoma skin cancers) featured at the top of the terms for the interactive models (*p* = 3.15 × 10^−21^, mEV = infinity, Table 2), in the model lagged by two years (*p* = 3.40 × 10^−3^, mEV = infinity, Appendix A), and also at four lags (*p* = 1.11 × 10^−41^, mEV = infinity, Table 4). In a marginal effects study, Δ8THC was noted to have an effect equal to that of AUD and above that of tobacco. Additionally of note was the very high strength of the association of these tumors with Δ8THC, as quantified by the 95% lower bound of the E-Value. In the continuous analysis, elevated mEVs ranged from 3.42 × 10^15^ to 27.14 (Appendix A) and in the additive panel model, from 2.64 × 10^14^ to 42.69 (Appendix A). In the interactive IPW panel models lagged to zero, two, and four years, 73.5%, 80.5%, and 97.9% of 49, 41, and 49 elevated E-Values were infinite with mEVs declining from infinity in each case to 56.10, 2.85 × 10^12^, and 2.31 × 10^123^, respectively.

### 4.2. Mechanisms

A discussion of the mechanisms of Δ8THC’s cellular actions is highly pertinent to the Hill criteria of causality under the biological plausibility clause [68]. Δ8THC is less well-studied in this respect than Δ9THC. Cells carry all the machinery of cannabinoid signal reception and transduction in the inner and outer membranes of mitochondria [17,20,21], which are in free and ready communication with the nucleus, and Δ8THC binds CB1R [2,5] and inhibits mitochondrial activity [17,18,19,20,21]. Mitochondria supply both small molecular epigenetic substrates and energy required for genome maintenance and stability and the epigenetic machinery. As was noted above, Δ8THC is known to inhibit DNA, RNA, and protein synthesis [5,6,36,37,38,39]. Extensions of such studies with Δ9THC showed that Δ9THC inhibited histone synthesis (in excess of 50%) [69,70], which necessarily leads to an open chromatin conformation and therefore has major effects on the availability of chromatin for transcription in a pro-oncogenic manner. Δ9THC has been shown to greatly alter DNA methylation in the sperm of mice, rats, and men [14,15,16,71,72], inducing changes that persist in altered brain and appetitive center function in subsequent generations [16,73,74], perturbations that have been shown to improve with cannabis abstinence [75]. Indeed, if one adds the length of the chromosomes (13,18, 21, X), recently shown to be affected by cannabis-induced trisomies/monosomy [27] to those affected in testicular carcinoma (1, 7, 8, 11, 12, 13, 18, 21, X, Y) [76] and commonly in acute lymphoid leukemia (4, 9, 10, 11, 22) [77], one arrives at the sizeable length of 1765 MB or 58.8% of the 3000 MB of the human genome directly impacted by chromosomal toxicity alone.

### 4.3. Mechanisms of Cannabinoid Carcinogenesis

The link between cancer and cannabinoids is complex and multifactorial and has recently been reviewed in several recently published works [10,14,15,16,27,31,32,34,35,41,42,43,71,72,75,78,79,80,81,82,83,84,85,86,87,88,89,90,91,92]. Our intention here is to introduce the way in which these general observations may relate to some specific cancer types. We also considered recent important epigenomic findings and mechanisms underlying chromosomal mis-segregation events and chromosomal breakage and translocation events that are known to be important oncogenic events underlying tumors such as acute myeloid and lymphoid leukemias and testicular cancer [25,76,77,93,94,95].

### 4.4. Recent DNA Methylation Studies

Of particular importance is the recent finding emerging from an epigenome wide association study (EWAS) of twenty cannabis dependent patients and twenty control subjects compared to each other both before and after an eleven week period of documented cannabis sobriety [75]. This study identified 810 hits relating to cancer-related terms (including “neoplasm”, “carcinogenesis”, “tumor”, “carcinoma”, “leukemia” and “lymphoma”) and mentioned specifically cancers of the haemopoietic system (leukemia, lymphoma, myeloma), breast, ovarian, colorectal, prostate, brain, pancreatic, thyroid, liver, melanoma, esophageal, and gastric and upper respiratory tract tumors. Indeed, the cancer signal was one of the strongest overall signals to emerge from this EWAS. The full implications of this profound and far-reaching result have yet to be fully explored. They have an obvious relevance and concordance with the results reported both in this study and in similar reports [27,79,80,81,96,97,98].

### 4.5. Chromosomal Structural Observations

#### 4.5.1. Shedding of Chromosomal Arms and Possible Breakage–Fusion–Bridge Cycles

Several classical studies have provided clear evidence of chromosomal breaks [9,10,13], end-to-end fusions [9,10], and chromosomal bridge formations in telophase [11,12,99] following cannabis exposure. Such findings suggest that the breakage–fusion–bridge cycle described by Barbara McClintock may be in operation [100,101] for all of the elements of this cycle have clearly been shown to be present. If such a positive feed-forward cycle were operating, it would explain the aggressive loss of 50–70 chromosomal arms from testicular tumor germ cells in the non-seminomatous germ cell tumors, which is well-established to be linked with cannabis [76]. It may be that careful experimental investigation of the mechanistic basis of this florid loss of chromosomal segments would be very informative.

#### 4.5.2. Errors of Meiosis and Mitosis

Hyperploidy and supernumerary chromosomal replication has been demonstrated following cannabis exposure in mammalian lymphocytes and oocytes [12,99]. Such a finding would be consistent with the one or two rounds of genomic doubling required in the biogenesis of testicular carcinogenesis [76].

Trisomies (of chromosomes 13, 18 and 21) and monosomies (of chromosome X, Turners syndrome) have been well-documented following cannabis exposure [44,46,47,81,102], where chromosomal mis-segregation is clearly a major feature of cannabis-related genotoxicity. Moreover, male disomy of chromosome X (Kleinfelters syndrome) has also been observed in association with cannabis exposure in the European data (manuscript submitted). Such strong epidemiological evidence of chromosomal mis-segregation is supported by the induction of lagging chromosomes following cannabis exposure [7,8,10,86,88,103] and by the well-known positive results of cannabis on testing in the micronucleus assay [104].

The mitotic spindle is comprised of microtubules that are polymers of tubulin [105]. Cannabis has been shown to inhibit tubulin synthesis [70]. Therefore, interference with the integrity of the mitotic spindle is one mechanism by which cannabinoids may disrupt cell division [105,106].

#### 4.5.3. Epigenomic Control of Chromosomal Centromeric Function

The binding of chromosomes to the mitotic spindle in mammals is mediated by the kinetochore, which is a large 90-protein complex that binds centromeric chromatin of each chromosome to 25–30 microtubules of the mitotic spindle [107]. Most of the arms of the mitotic spindle are actually incomplete and the spindle therefore is actually two half-spindles that are linked by complete rays, which course from one controller to the other. Chromosomes are bound to the growing plus ends of the half spindle rays. The chromosomes separate at anaphase due to the pulling forces exerted by dynein–dynactin molecular motors, which retract the chromosomes toward the minus ends of the microtubules and the new pronuclear poles [108,109,110,111,112,113].

The histones occurring in centromeric chromatin are specialized and mark this section of the chromatin uniquely. A group of CENP-A variants of histone 3 are some of the most important of these modified histones [114]. H3-CENP-As have been widely conserved across plant and animal phyla [115]. Pericentromeric histones are richly decorated in post-translational epigenomic modifications (PTM). SUMOylation is the addition of Small Ubiquitin-like MOdifier (SUMO) proteins and is one of the most important post-translational modifications, which becomes a key step in the formation of the rich and complex chains of PTMs including (methylation, acetylation, tyrosinylation, sulfation, phosphorylation, ubiquitylation, etc.), which then control kinetochore function combinatorially [116]. Specific PTMs that are controlled by sumoylation include H3K4me1, H3K4me2, H3K4me3, H3- and H4-acetylation, and H2BK123 ubiquitination [116]. Thus, histone sumoylation acts as a key functional switch that controls the kinetochore function and its downstream signaling to release the Spindle Associated Checkpoint (SAC), which controls the anaphase segregation of chromosomes [107].

Therefore, the observation that this histone sumoylation switch is powerfully controlled by Δ9THC is of great relevance to the issue of chromosomal mis-segregation during anaphase and cytokinesis subsequent to cannabis exposure [117]. Ubiquitin-like-specific protease 2 (ulp2) is inhibited by Δ9THC, which blocks desumoylation and thereby disrupts the integrity of the sumoylation code [116]. Administration of Δ9THC to dividing cells thus causes major disruptions of the kinetochore signaling to the SAC and leads to errors of chromosomal segregation [116]. Δ9THC also directly affects murine double minute 2 (mdm2) and SUMO-1 protein major binding partners of P53, which is well-known as the classical “guardian of the genome” [117]. Through interference with mdm2 and SUMO-1, Δ9THC activates P53 directly. P53 in cannabis-exposed cells will thus be activated both canonically via the induction of DNA single- and double-stranded breaks and also via its protein interactome.

Thus, pericentromeric chromatin is regulated by a series of sophisticated epigenomic mechanisms through the critical stages of attachment to the mitotic spindle and chromosomal segregation, and cannabinoids seriously disrupt this complex post-translational interacting signalome. Since this centromeric epigenomic nucleosomal mechanism controls chromosomal segregation and may also be involved in chromosomal counting and identification systems, it becomes apparent that the disturbance of kinetochore function plays a pivotal and central role in both the induction of chromosomal mis-segregation errors, and likely hyperploidy and their downstream sequelae.

#### 4.5.4. Epigenomic Impacts on DNA Breakage Sites

Both single- and double- stranded breaks (SSB and DSB) are a common feature of cell karyotype studies after cannabis exposure [7,8,9,10,12,13,99]. It therefore becomes important in the present context to note that the epigenome plays an often determinative role in influencing or selecting the site of DNA breakage generally [118,119,120,121,122,123,124,125,126,127,128,129,130,131], during meiotic crossing over [132,133,134,135,136,137,138,139], in the immune gene hypervariable region [140,141,142,143,144,145,146], and in oncogenic pathways [120,123,124,147,148,149,150,151,152,153,154].

The presence of acute lymphoid leukemia (ALL) listed in Table 2 and Table 4 in the present study is significant. This disorder is known to commonly represent end-to-end translocations and fusions between several chromosomes (such as 4 and 11, 9 and 22, 4 and 10) [77]. As well as in the present results, ALL was also recently shown to be elevated in association with population-wide cannabis exposure. The chromosomal and genomic lesions of this tumor imply increased chromosomal double-stranded breaks, and anomalous repair, resulting in the chromosomal translocation landscape described [77].

#### 4.5.5. Cannabinoids Deliver Multiple Carcinogenic Insults

The multi-hit hypothesis is one of the common models of carcinogenesis and involves multiple genotoxic or epigenotoxic hits to the genome, which result in genomic instability [94,155,156,157,158]. As cannabinoids can deliver both double- and single-stranded DNA breaks and cause the disruption of key kinetochore functions (hypoploidy [11], hyperploidy [12,99], and chromosomal mis-segregation [27,44,47,102]), it is clear that significant cannabinoid exposure can deliver multi-point genomic hits in themselves. This also explains the dramatic abnormalities of cell karyotype from the minimal cannabis exposure (just a few puffs) observed in many classical cell morphology studies [7,8,9,11,12,86,88,99].

Testicular carcinoma was noted to be elevated in Table 1. This tumor almost invariably involves the development of an isochromosome 12p. Its presence is explained by the concept of presumptive pericentromeric chromatin dysregulation as the dysregulated pericentromeric epigenome presumably facilitates the aberrant scission of the chromosome at the centromere, forming the isochromosome through the interactions between the epigenome and the genome as described above [118,119,120,121,122,123,124,125,126,127,128,129,130,131]. The presence of KIT and KRAS (and to a lesser extent NRAS) on chromosome 12 then confers a growth advantage on the mutant clone, and malignant tumorigenesis is the end result of this process continued in the context of the gross re-sculpting of the chromosomal landscape by repeated cycles of the breakage–fusion–bridge cycle across multiple cell divisions that accumulate over time.

### 4.6. Causal Inference

Qualitative Causal Inference. The qualitative criteria required of causal relationships were set out by Hill in 1965 [68] and it is noted that this study fulfills all these criteria, except those that require replication elsewhere, which is unavoidable in an initial study. Hence, the present results demonstrate the strength of association, specificity, temporality, coherence with known data, biological plausibility, a biological dose–response gradient, and experimental confirmation. In light of its pioneering nature, we were not able to demonstrate consistency amongst studies or analogy with similar situations elsewhere.

Quantitative Causal Inference. The use of inverse probability weighting in all panel models has the effects of making all groups in an observational study pseudo-randomized and allows for causal inferences to be drawn from observational data. Its results have recently been checked against later randomized control trials and these effects have been confirmed. Similarly E-Values, or the expected values, quantitates the degree of co-association required of some hypothetical external confounder variable with both the exposure and the outcome in order to explain away the observed effect. In the present study, the many very elevated mEVs at infinity preclude external explanations on the quantitative criteria, which is commonly used to indicate causal pathways above 1.25 [64]. As this study combined both inverse probability weighting and mEVs, it becomes a powerful framework within which to consider causal relationships.

### 4.7. Generalizability

We feel that these results are generalizable for the quantitative reasons noted above because they fulfill the quantitative criteria of causal inference. Not only do these results fulfill the quantitative criteria, but they also fulfill most of the quantitative criteria of Hill [68] including the strength of association, specificity, temporality, coherence with other known data, biological plausibility, biological dose-response gradient, and experimental confirmation. As this is the first study of this type to our knowledge, the only criteria that is not met is that results have not been replicated elsewhere as this is the first study of this type to our knowledge. As the relationship appears to be defined causally, the expectation is that this link would be shown wherever data of sufficient quality exists.

### 4.8. Strengths and Limitations

This study has a number of strengths and limitations. Its strengths include that it uses large national datasets with high response rates, that the analytical plan is relatively simple (continuous and categorical bivariate and multivariate) but also powerful including multiple simultaneous model processing (via purrr) and liberal use of the tools of causal inference, particularly inverse probability weighting and E-Values. Its limitations are that we had to rely on estimates of the state Δ8THC exposure as the actual data are not available, however, this is a common practice in such studies of the cancer epidemiology of individual cannabinoids [27,28,159]. Like many epidemiological studies, individual participant level data were not available to the present investigators. Whilst cancer-specific risk factors were not studied (such as hormonal exposure, body mass index, and regular exercise), the very high E-Values reported in this study (see Table 2 and Table 4) indicate that the inclusion of such covariates would not greatly perturb the main results reported herein.

## 5. Conclusions

In conclusion, this study demonstrated that Δ8THC exposure may be directly linked to eight cancer types in bivariate testing and as many as 18 of the 34 tumor types assessed on inverse panel regression persisted after full multivariable adjustment. The effect sizes reported herein were remarkably strong, often ranging up to infinity for the minimum E-Values of significant terms in multivariable models. The overall marginal effect shown was around 8.6/100,000, a figure comparable to alcohol and above that for tobacco. These results are consistent both with other recent reports of cannabinoid-related carcinogenesis in adults and children and a basic science genotoxic and epigenotoxic literature, which has long documented the genotoxicity and epigenotoxicity of multiple cannabinoids. Documented relationships fulfill both quantitative and qualitative criteria for causality. These results sound a strong note of warning that the present commercially driven renewed interest and popularity of Δ8THC in the USA may portend a major epidemic in years and generations to come of genotoxic including cancerogenic and other outcomes not only in the deliberately exposed, but also potentially spilling over into the food chain with effects on general communities, systemic and inheritable epigenomic aging, and having downstream outcomes for several generations to come.

## Figures and Tables

**Figure 1 ijerph-19-07726-f001:**
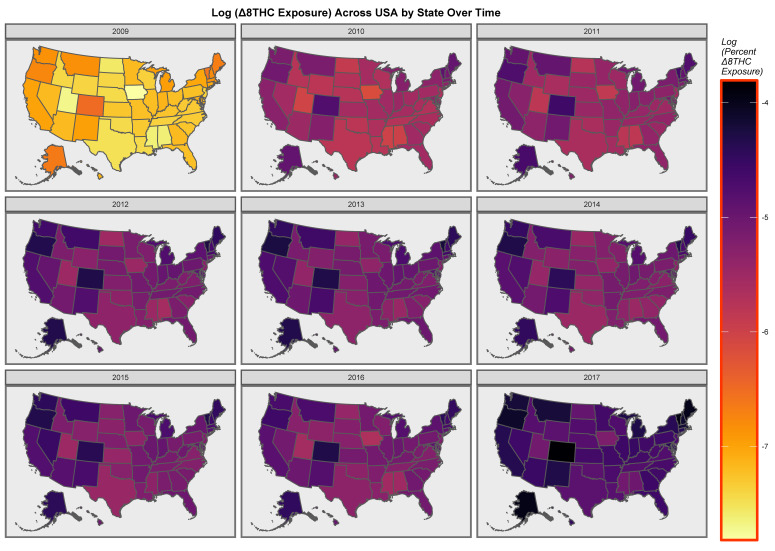
Map of the log (Δ8THC estimates) across the USA over time.

**Figure 2 ijerph-19-07726-f002:**
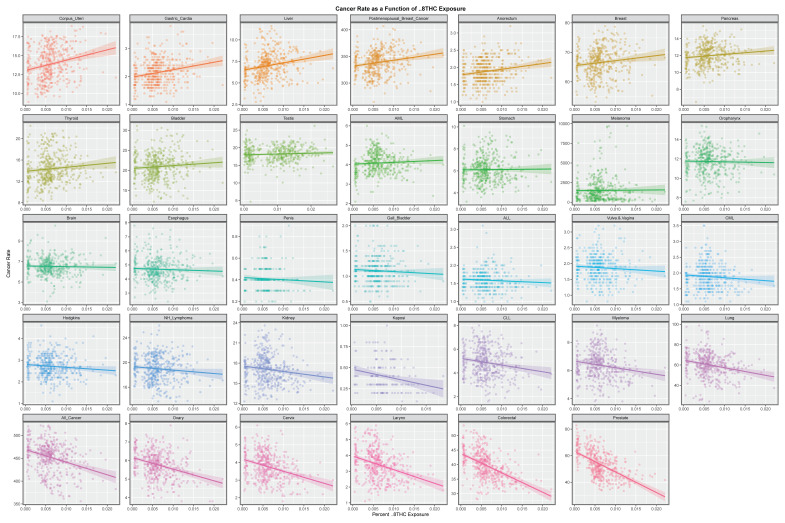
The bivariate continuous relationships of the Δ8THC rates by cancer type.

**Figure 3 ijerph-19-07726-f003:**
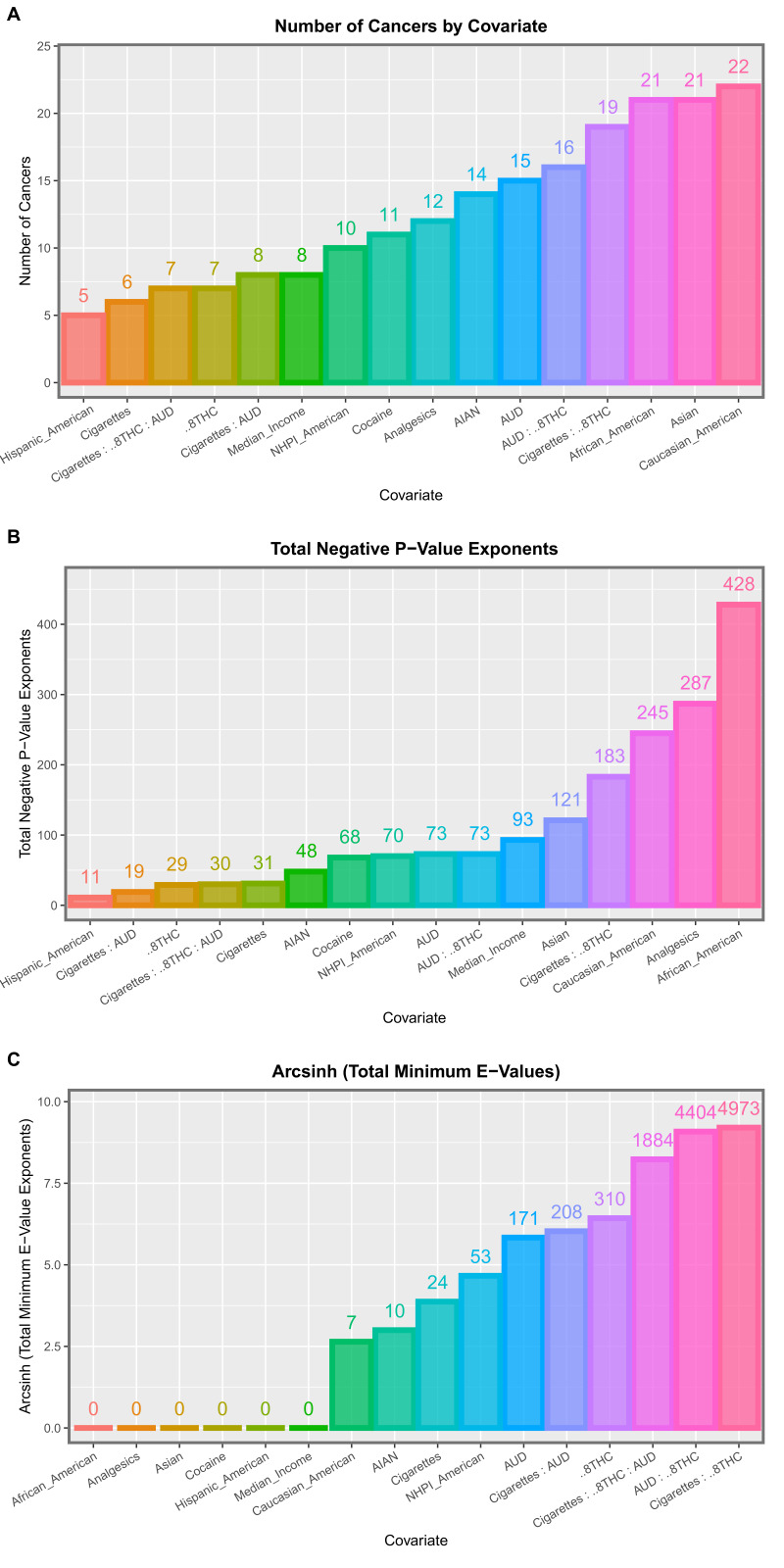
Summary graphs for the interactive multivariable panel models. Please see text for details.

**Figure 4 ijerph-19-07726-f004:**
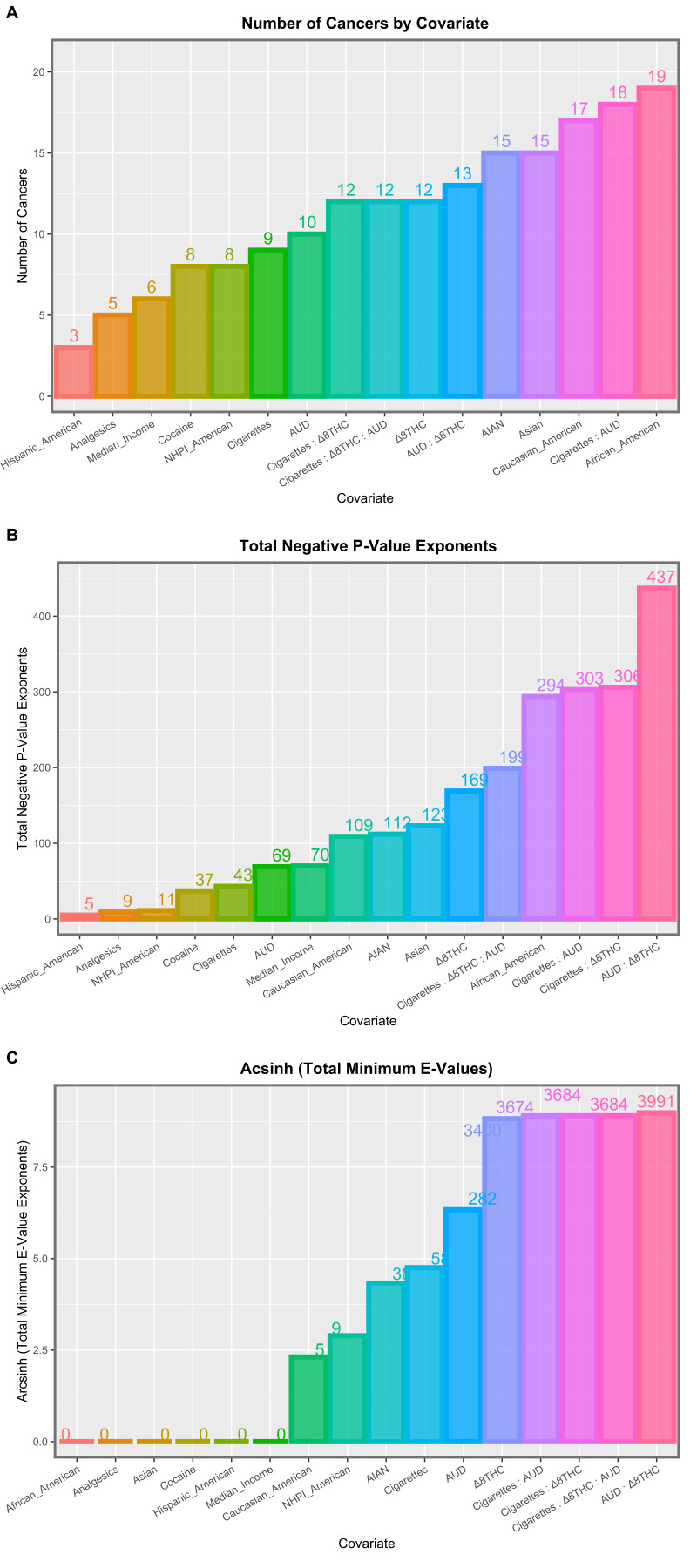
Summary graphs for the interactive multivariable panel models at four years of lag.

**Figure 5 ijerph-19-07726-f005:**
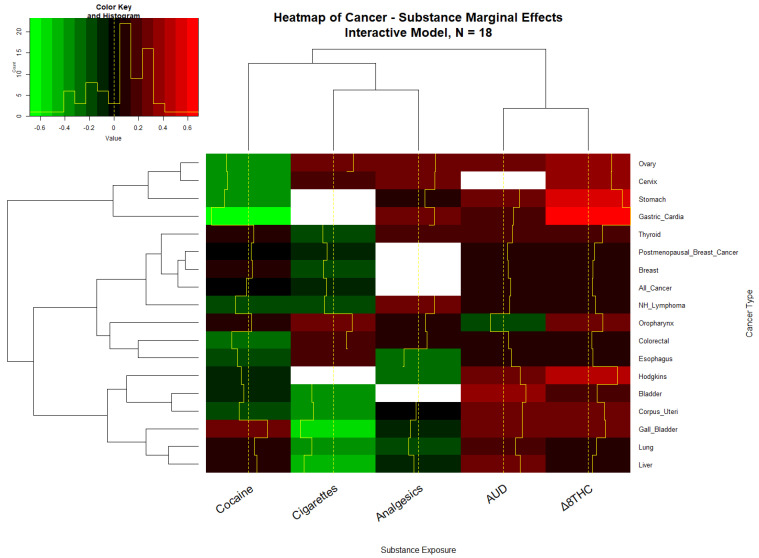
The heatmap of the cancer–substance marginal effects, Interactive IPW model.

**Table 1 ijerph-19-07726-t001:** The continuous bivariate relationships of Δ8THC.

Cancer	Cases in Quintile 5	Non-Cases in Quintile 5	Cases in Quintile 1	Non-Cases in Quintile 1	R.R.	R.R. (Lower C.I.)	R.R. (Upper C.I.)	A.F.E.	A.F.E. (Lower C.I.)	A.F.E. (Upper C.I.)	P.A.R.	P.A.R. (Lower C.I.)	P.A.R. (Upper C.I.)	Chi. Squared	*p*-Value	E-Value Estimate	E-Value (Lower C.I.)
Melanoma	100,207	384,314,333	115,471	540,452,122	1.2203	1.2100	1.2307	0.1805	0.1736	0.1874	0.0839	0.0803	0.0875	2134.601	0	1.74	1.71
Liver	49,147	384,365,393	62,900	540,504,693	1.0987	1.0859	1.1118	0.0899	0.0791	0.1005	0.0394	0.0344	0.0444	244.865	1.71 × 10^−55^	1.43	1.39
Corpus_Uteri	78,333	384,336,207	100,741	540,466,852	1.0934	1.0833	1.1037	0.0854	0.0769	0.0939	0.0374	0.0334	0.0413	351.830	8.46 × 10^−79^	1.41	1.38
Anorectum	10,462	384,404,078	13,537	540,554,056	1.0868	1.0594	1.1149	0.0799	0.0561	0.1030	0.0348	0.0240	0.0455	40.895	8.03 × 10^−11^	1.39	1.31
Thyroid	67,875	384,346,665	89,788	540,477,805	1.0630	1.0525	1.0737	0.0593	0.0499	0.0686	0.0255	0.0213	0.0297	144.443	1.42 × 10^−33^	1.32	1.29
Gastric_Cardia	144,970	384,269,570	193,373	540,374,220	1.0542	1.0471	1.0614	0.0514	0.0450	0.0579	0.0220	0.0192	0.0249	231.159	1.67 × 10^−52^	1.29	1.27
AML	23,937	384,390,603	31,708	540,535,885	1.0616	1.0439	1.0795	0.0580	0.0421	0.0737	0.0250	0.0179	0.0320	48.720	1.48 × 10^−12^	1.32	1.26
Pancreas	67,490	384,347,050	90,446	540,477,147	1.0493	1.0389	1.0598	0.0470	0.0374	0.0564	0.0201	0.0159	0.0242	89.545	1.50 × 10^−21^	1.28	1.24
Postmenopausal Breast Cancer	249,834	384,164,706	337,900	540,229,693	1.0397	1.0344	1.0451	0.0382	0.0332	0.0432	0.0162	0.0141	0.0184	218.017	1.22 × 10^−49^	1.24	1.22
Breast	337,544	384,076,996	462,078	540,105,515	1.0272	1.0227	1.0318	0.0265	0.0222	0.0308	0.0112	0.0093	0.0130	140.859	8.63 × 10^−33^	1.19	1.17
Testis	52,627	1,726,611,593	19,713	670,166,199	1.0362	1.0194	1.0533	0.0349	0.0190	0.0506	0.0254	0.0137	0.0369	18.136	1.03 × 10^−5^	1.23	1.16
Oropharynx	63,178	384,351,362	87,440	540,480,153	1.0160	1.0057	1.0265	0.0158	0.0057	0.0258	0.0066	0.0023	0.0109	9.276	0.0012	1.14	1.08
Myeloma	31,335	384,383,205	44,787	540,522,806	0.9838	0.9697	0.9982	−0.0164	−0.0312	−0.0019	−0.0068	−0.0128	−0.0008	4.889	0.0135	1.15	-
Bladder	103,627	384,310,913	148,415	540,419,178	0.9819	0.9741	0.9897	−0.0185	−0.0266	−0.0104	−0.0076	−0.0109	−0.0043	20.478	3.02 × 10^−6^	1.16	-
Vulva.&.Vagina		375,122,812	20,981	536,846,206	0.9803	0.9597	1.0013	−0.0201	−0.0420	0.0013	−0.0082	−0.0169	0.0005	3.370	0.0332	1.16	-
CML	9482	373,925,516	13,840	532,699,861	0.9760	0.9509	1.0019	−0.0246	−0.0517	0.0019	−0.0100	−0.0208	0.0007	3.313	0.0344	1.18	-
Penis	2283	312,646,824	3380	451,235,667	0.9749	0.9244	1.0280	−0.0258	−0.0817	0.0272	−0.0104	−0.0323	0.0110	0.884	0.1736	1.19	-
Esophagus	23,815	382,573,677	34,228	535,139,818	0.9732	0.9573	0.9895	−0.0275	−0.0446	−0.0106	−0.0113	−0.0182	−0.0044	10.330	6.54 × 10^−4^	1.20	-
ALL	6865	344,103,689	9149	445,794,343	0.9721	0.9422	1.0030	−0.0287	−0.0614	0.0030	−0.0123	−0.0260	0.0012	3.140	0.0382	1.20	-
Brain	29,472	384,385,068	42,767	540,524,826	0.9691	0.9548	0.9835	−0.0319	−0.0474	−0.0167	−0.0130	−0.0192	−0.0069	17.236	1.65 × 10^−5^	1.21	-
Kidney	82,655	384,331,885	120,241	540,447,352	0.9666	0.9581	0.9752	−0.0345	−0.0437	−0.0254	−0.0141	−0.0177	−0.0104	56.391	2.97 × 10^−14^	1.22	-
NH_Lymphoma	93,104	384,321,436	137,276	540,430,317	0.9537	0.9458	0.9617	−0.0485	−0.0573	−0.0398	−0.0196	−0.0230	−0.0162	124.585	3.14 × 10^−29^	1.27	-
Gall_Bladder	14,123	370,777,042	21,633	534,521,894	0.9412	0.9214	0.9613	−0.0625	−0.0853	−0.0402	−0.0247	−0.0333	−0.0161	31.433	1.03 × 10^−8^	1.32	-
Stomach	32,140	384,382,400	48,205	540,519,388	0.9376	0.9244	0.9509	−0.0666	−0.0817	−0.0516	−0.0266	−0.0324	−0.0209	80.166	1.72 × 10^−19^	1.33	-
All_Cancer	2,151,597	382,262,943	3,244,348	537,323,245	0.9326	0.9310	0.9342	−0.0723	−0.0741	−0.0705	−0.0288	−0.0295	−0.0281	6343.397	0	1.35	-
Hodgkins	10,210	384,404,330	15,405	540,552,188	0.9320	0.9090	0.9556	−0.0730	−0.1001	−0.0465	−0.0291	−0.0394	−0.0189	30.469	1.70 × 10^−8^	1.35	-
CLL	26,057	384,388,483	39,705	540,527,888	0.9228	0.9085	0.9374	−0.0836	−0.1007	−0.0668	−0.0331	−0.0395	−0.0267	101.487	3.60 × 10^−24^	1.38	-
Ovary	26,688	384,387,852	43,321	540,524,272	0.8663	0.8532	0.8796	−0.1543	−0.1721	−0.1369	−0.0588	−0.0650	−0.0527	340.797	2.14 × 10^−76^	1.58	-
Lung	272,701	384,141,839	452,339	540,115,254	0.8478	0.8437	0.8518	−0.1796	−0.1852	−0.1740	−0.0675	−0.0695	−0.0656	4654.942	0	1.64	-
Cervix	14,961	384,399,579	25,037	540,542,556	0.8403	0.8234	0.8575	−0.1901	−0.2144	−0.1662	−0.0711	−0.0792	−0.0630	284.293	4.36 × 10^−64^	1.67	-
Colorectal	171,103	384,243,437	286,861	540,280,732	0.8388	0.8338	0.8438	−0.1922	−0.1994	−0.1851	−0.0718	−0.0742	−0.0694	3323.813	0	1.67	-
Larynx	14,027	384,400,513	26,349	540,541,244	0.7486	0.7334	0.7641	−0.3358	−0.3635	−0.3087	−0.1167	−0.1246	−0.1087	772.855	2.15 × 10^−170^	2.01	-
Prostate	218,368	384,196,172	413,423	540,154,170	0.7428	0.7389	0.7466	−0.3463	−0.3533	−0.3394	−0.1197	−0.1217	−0.1177	12739.787	0	2.03	-
Kaposi	1270	177,333,272	1589	143,565,460	0.6471	0.6010	0.6966	−0.5455	−0.6638	−0.4356	−0.2423	−0.2837	−0.2023	135.891	1.05 × 10^−31^	2.46	-

**Table 2 ijerph-19-07726-t002:** The positive significant terms including Δ8THC from the interactive panel model.

Cancer	Term	Estimate	Std. Error	t-Statistic	S.D.	Adj. R. Squared	*p*-Value	E-Value Estimate	E-Value (Lower C.I.)
All_Cancer	cigmon: Δ8THC	798.48	80.06	9.97	0.24	0.27	3.15 × 10^−21^	Infinity	Infinity
Lung	cigmon: Δ8THC	2052.34	228.44	8.98	0.68	0.31	8.11 × 10^−18^	Infinity	Infinity
Stomach	cigmon: Δ8THC	3221.86	395.94	8.14	1.17	0.02	4.33 × 10^−15^	Infinity	Infinity
Gall_Bladder	cigmon: Δ8THC	2700.28	340.42	7.93	0.99	−0.04	2.32 × 10^−14^	Infinity	Infinity
NH_Lymphoma	cigmon: Δ8THC	1326.92	170.89	7.76	0.51	0.01	5.96 × 10^−14^	Infinity	Infinity
Kidney	cigmon: Δ8THC	1253.70	162.80	7.70	0.48	0.09	9.27 × 10^−14^	Infinity	Infinity
CML	cigmon: Δ8THC	2204.88	308.65	7.14	0.88	0.00	4.42 × 10^−12^	Infinity	Infinity
Thyroid	cigmon: Δ8THC	1681.77	242.80	6.93	0.72	−0.01	1.57 × 10^−11^	Infinity	Infinity
Vulva.&.Vagina	cigmon: Δ8THC	2190.07	328.80	6.66	0.97	0.04	8.29 × 10^−11^	Infinity	Infinity
Bladder	cigmon: Δ8THC	1224.02	190.07	6.44	0.56	0.22	3.18 × 10^−10^	Infinity	Infinity
Pancreas	cigmon: Δ8THC	1553.31	245.48	6.33	0.73	0.06	6.22 × 10^−10^	Infinity	Infinity
Cervix	cigmon: Δ8THC	1859.44	320.08	5.81	0.95	0.08	1.22 × 10^−8^	Infinity	Infinity
NH_Lymphoma	Δ8THC: AUD	2282.42	393.82	5.80	0.51	0.01	1.31 × 10^−8^	Infinity	Infinity
Stomach	Δ8THC: AUD	5230.42	912.47	5.73	1.17	0.02	1.86 × 10^−8^	Infinity	Infinity
Prostate	cigmon: Δ8THC: AUD	8926.85	1562.45	5.71	0.39	0.26	2.06 × 10^−8^	Infinity	Infinity
All_Cancer	Δ8THC: AUD	1052.81	184.49	5.71	0.24	0.27	2.14 × 10^−8^	Infinity	Infinity
Lung	Δ8THC: AUD	2939.06	526.46	5.58	0.68	0.31	4.19 × 10^−8^	Infinity	Infinity
Corpus_Uteri	cigmon: Δ8THC	1158.96	213.51	5.43	0.63	0.13	9.50 × 10^−8^	Infinity	Infinity
Melanoma	cigmon: Δ8THC: AUD	56,176.95	10656.53	5.27	2.64	0.04	2.14 × 10^−7^	Infinity	Infinity
ALL	cigmon: Δ8THC: AUD	34,385.14	6634.60	5.18	1.53	−0.04	3.67 × 10^−7^	Infinity	Infinity
Colorectal	Δ8THC: AUD	1353.86	266.99	5.07	0.34	0.15	5.88 × 10^−7^	Infinity	Infinity
CML	Δ8THC: AUD	3523.42	702.47	5.02	0.88	0.00	8.01 × 10^−7^	Infinity	Infinity
Colorectal	cigmon: Δ8THC	553.21	115.85	4.78	0.34	0.15	2.46 × 10^−6^	Infinity	Infinity

**Table 3 ijerph-19-07726-t003:** The summary of all terms from Δ8THC from the interactive panel model.

Covariate	Number of Cancers	Total Negative Exponent of *p*-Value	Total Negative Exponent of Lower E-Value Bound
AIAN American	14	48	10
AUD	15	73	171
Cigarettes	6	31	24
Cigarettes: AUD	8	19	208
Cigarettes: Δ8THC	19	183	4973
Cigarettes: Δ8THC: AUD	7	30	1884
Δ8THC	7	29	310
AUD: Δ8THC	16	73	4404
Analgesics	12	287	0
Asian American	21	121	0
African_American	21	428	0
Cocaine	11	68	0
Hispanic_American	5	11	0
Median Income	8	93	0
NHPI_American	10	70	53
Caucasian American	22	245	7

**Table 4 ijerph-19-07726-t004:** Significant terms including Δ8THC from the interactive panel model at four years lag.

Cancer	Term	Estimate	Std. Error	t-Statistic	S.D.	Adj. R. Squared	*p*-Value	E-Value Estimate	E-Value (Lower C.I.)
Kidney	lag(cigmon, 4): lag(Δ8THC, 4): lag(AUD, 4)	20,775.83	729.67	28.47	0.35	−0.03	8.04 × 10^−78^	Infinity	Infinity
AML	lag(Δ8THC, 4): lag(AUD, 4)	37,051.70	1513.53	24.48	0.72	−0.06	2.36 × 10^−66^	Infinity	Infinity
Pancreas	lag(Δ8THC, 4): lag(AUD, 4)	30,354.71	1244.74	24.39	0.60	−0.06	4.51 × 10^−66^	Infinity	Infinity
Kidney	lag(cigmon, 4): lag(Δ8THC, 4)	6008.51	278.08	21.61	0.35	−0.03	1.44 × 10^−57^	Infinity	Infinity
AML	lag(cigmon, 4): lag(Δ8THC, 4)	11,523.47	576.82	19.98	0.72	−0.06	2.09 × 10^−52^	Infinity	Infinity
Larynx	lag(Δ8THC, 4): lag(AUD, 4)	42,033.01	2140.10	19.64	1.02	0.12	2.54 × 10^−51^	Infinity	Infinity
Brain	lag(cigmon, 4): lag(Δ8THC, 4): lag(AUD, 4)	103,777.35	5628.18	18.44	0.63	−0.06	2.01 × 10^−47^	Infinity	Infinity
Pancreas	lag(cigmon, 4): lag(Δ8THC, 4)	8433.65	474.38	17.78	0.60	−0.06	2.93 × 10^−45^	Infinity	Infinity
Vulva.&.Vagina	lag(cigmon, 4): lag(Δ8THC, 4)	13,904.77	815.61	17.05	1.02	0.12	7.47 × 10^−43^	Infinity	Infinity
All_Cancer	lag(Δ8THC, 4): lag(AUD, 4)	4404.42	263.84	16.69	0.13	0.11	1.12 × 10^−41^	Infinity	Infinity
Brain	lag(Δ8THC, 4)	1766.26	115.19	15.33	0.63	−0.06	3.70 × 10^−37^	Infinity	Infinity
Liver	lag(Δ8THC, 4): lag(AUD, 4)	15,683.99	1078.13	14.55	0.52	0.22	1.52 × 10^−34^	Infinity	Infinity
ALL	lag(cigmon, 4): lag(Δ8THC, 4): lag(AUD, 4)	80,386.15	6115.36	13.14	0.66	−0.08	3.84 × 10^−28^	Infinity	Infinity
Liver	lag(cigmon, 4): lag(Δ8THC, 4)	4887.56	410.89	11.90	0.52	0.22	8.25 × 10^−26^	Infinity	Infinity
ALL	lag(Δ8THC, 4)	1491.45	125.55	11.88	0.66	−0.08	1.96 × 10^−24^	Infinity	Infinity
Oropharynx	lag(cigmon, 4): lag(Δ8THC, 4): lag(AUD, 4)	42,596.04	3835.11	11.11	0.43	0.27	2.79 × 10^−23^	Infinity	Infinity
EsophLagus	lag(cigmon, 4): lag(Δ8THC, 4): lag(AUD, 4)	79,727.71	7246.10	11.00	0.80	−0.06	8.94 × 10^−23^	Infinity	Infinity
All_Cancer	lag(cigmon, 4): lag(Δ8THC, 4)	1092.50	100.55	10.87	0.13	0.11	1.63 × 10^−22^	Infinity	Infinity
CLL	lag(Δ8THC, 4): lag(AUD, 4)	16,144.24	1490.78	10.83	0.71	0.09	2.11 × 10^−22^	Infinity	Infinity
NH_Lymphoma	lag(Δ8THC, 4): lag(AUD, 4)	11,559.70	1096.50	10.54	0.52	−0.03	1.68 × 10^−21^	Infinity	Infinity
EsophLagus	lag(Δ8THC, 4)	1551.04	148.10	10.47	0.80	−0.06	3.93 × 10^−21^	Infinity	Infinity
Oropharynx	lag(Δ8THC, 4)	765.23	78.49	9.75	0.43	0.27	4.72 × 10^−19^	Infinity	Infinity
Myeloma	lag(cigmon, 4): lag(Δ8THC, 4): lag(AUD, 4)	51,756.76	5401.86	9.58	0.61	0.02	1.52 × 10^−18^	Infinity	Infinity
Postmenopausal Breast Cancer	lag(cigmon, 4): lag(Δ8THC, 4): lag(AUD, 4)	23,566.70	2616.35	9.01	0.29	0.10	7.76 × 10^−17^	Infinity	Infinity
Postmenopausal Breast Cancer	lag(Δ8THC, 4)	461.33	53.55	8.61	0.29	0.10	1.07 × 10^−15^	Infinity	Infinity
Bladder	lag(Δ8THC, 4): lag(AUD, 4)	11,273.58	1317.42	8.56	0.63	0.06	1.57 × 10^−15^	Infinity	Infinity
Anorectum	lag(Δ8THC, 4): lag(AUD, 4)	16,569.38	1977.38	8.38	0.95	−0.02	5.04 × 10^−15^	Infinity	Infinity
CLL	lag(cigmon, 4): lag(Δ8THC, 4)	4683.01	568.15	8.24	0.71	0.09	1.23 × 10^−14^	Infinity	Infinity
Lung	lag(Δ8THC, 4): lag(AUD, 4)	9636.37	1178.24	8.18	0.56	0.10	1.85 × 10^−14^	Infinity	Infinity
Myeloma	lag(Δ8THC, 4)	856.96	110.56	7.75	0.61	0.02	2.81 × 10^−13^	Infinity	Infinity
Thyroid	lag(Δ8THC, 4): lag(AUD, 4)	6736.53	885.15	7.61	0.42	0.18	6.73 × 10^−13^	Infinity	Infinity
Bladder	lag(cigmon, 4): lag(Δ8THC, 4)	3651.37	502.08	7.27	0.63	0.06	5.33 × 10^−12^	Infinity	Infinity
Anorectum	lag(cigmon, 4): lag(Δ8THC, 4)	4954.17	753.59	6.57	0.95	−0.02	3.18 × 10^−^10	Infinity	Infinity
Prostate	lag(cigmon, 4): lag(Δ8THC, 4): lag(AUD, 4)	16,430.44	2513.72	6.54	0.28	0.15	3.94 × 10^−^10	Infinity	Infinity
NH_Lymphoma	lag(cigmon, 4): lag(Δ8THC, 4)	2661.93	417.88	6.37	0.52	−0.03	9.98 × 10^−^10	Infinity	Infinity
Ovary	lag(cigmon, 4): lag(Δ8THC, 4): lag(AUD, 4)	41,343.79	6533.78	6.33	0.73	−0.02	1.26 × 10^−9^	Infinity	Infinity
Ovary	lag(Δ8THC, 4)	806.38	133.73	6.03	0.73	−0.02	6.38 × 10^−9^	Infinity	Infinity
Thyroid	lag(cigmon, 4): lag(Δ8THC, 4)	1993.67	337.34	5.91	0.42	0.18	1.21 × 10^−8^	Infinity	Infinity
Breast	lag(cigmon, 4): lag(Δ8THC, 4): lag(AUD, 4)	10,735.82	1828.95	5.87	0.21	0.30	1.49 × 10^−8^	Infinity	Infinity
Prostate	lag(Δ8THC, 4)	299.18	51.45	5.82	0.28	0.15	1.99 × 10^−8^	Infinity	Infinity
Breast	lag(Δ8THC, 4)	210.27	37.43	5.62	0.21	0.30	5.51 × 10^−8^	Infinity	Infinity
Hodgkins	lag(Δ8THC, 4)	1216.18	217.41	5.59	1.19	0.03	6.20 × 10^−8^	Infinity	Infinity
Lung	lag(cigmon, 4): lag(Δ8THC, 4)	2417.46	449.04	5.38	0.56	0.10	1.78 × 10^−7^	Infinity	Infinity
Hodgkins	lag(cigmon, 4): lag(Δ8THC, 4): lag(AUD, 4)	55,559.09	10,622.37	5.23	1.19	0.03	3.76 × 10^−7^	Infinity	Infinity
Vulva.&.Vagina	lag(cigmon, 4): lag(Δ8THC, 4): lag(AUD, 4)	61,495.75	13,454.10	4.57	1.51	−0.01	8.25 × 10^−6^	Infinity	Infinity
Stomach	lag(cigmon, 4): lag(Δ8THC, 4): lag(AUD, 4)	33,237.10	7879.98	4.22	0.89	−0.03	3.53 × 10^−5^	Infinity	Infinity
Stomach	lag(Δ8THC, 4)	669.75	161.28	4.15	0.89	−0.03	4.61 × 10^−5^	Infinity	Infinity
Corpus_Uteri	Caucasian American	2187.58	1013.65	2.16	0.48	−0.01	3.19 × 10^−2^	Infinity	Infinity
Vulva.&.Vagina	lag(Δ8THC, 4)	1006.92	275.18	3.66	1.51	−0.01	3.19 × 10^−4^	Infinity	2.31 × 10^123^

**Table 5 ijerph-19-07726-t005:** A summary of all terms from Δ8THC from the interactive panel model at four years lag.

Covariate	Number of Cancers	Total Negative Exponent of *p*-Value	Total Negative Exponent of Lower E-Value Bound
AIAN American	15	112	38
AUD	10	69	282
Cigarettes	9	43	58
Cigarettes: AUD	18	303	3674
Cigarettes: Δ8THC	12	306	3684
Cigarettes: Δ8THC: AUD	12	199	3684
Δ8THC	12	169	3400
AUD: Δ8THC	13	437	3991
Analgesics	5	9	0
Asian American	15	123	0
African_American	19	294	0
Cocaine	8	37	0
Hispanic_American	3	5	0
Median Income	6	70	0
NHPI_American	8	11	9
Caucasian American	17	109	5

## Data Availability

All data generated or analyzed during this study are included in this published article and its Appendix A files. Data, along with the relevant R code, have been made publicly available on the Mendeley Database Repository and can be accessed from this URL, doi:10.17632/vd6mt5r5jm.1.

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
