# Peer review of "Epidemiology of Δ8THC-Related Carcinogenesis in USA: A Panel Regression and Causal Inferential Study"

_ijerph, 2022, doi:10.3390/ijerph19137726_

Round 1
Reviewer 1 Report
The authors represent the effect of delta8THC on genotoxicity and epigenotoxicity.
Authors need to consider the General Considerations - especially Research manuscript sections and more clearly separate these sections from other text. The abstract should be reorganized as consistent text without section names. Section Materials and Methods should be written in a more orderly manner. The tables and figures are a little crowdy – if possible keep them clear (example: fonts) and self-explainable. In the discussion, some sections (“literature review”) are in my opinion more suitable for the introduction section - but could stay in the discussion section with comparison or linkage to the study results (please add where relevant).
I wonder if statistics were performed by a professional statistician or statistical department.
Author Response
Response to Reviewers
Reviewer 1
The authors represent the effect of delta8THC on genotoxicity and epigenotoxicity.
Authors need to consider the General Considerations - especially Research manuscript sections and more clearly separate these sections from other text. The abstract should be reorganized as consistent text without section names.
Response.
This has now been done.
Section Materials and Methods should be written in a more orderly manner. The tables and figures are a little crowdy – if possible keep them clear (example: fonts) and self-explainable.
Response.
We agree with this remark.
We feel that the Tables will reproduce better in the final published format.
The Figures have all been expanded.
In this format they are easier to read.
Figures 3 and 4 have been changed to portrait layout.
This improved their readability greatly.
High resolution pdf files of Figures 1-5 are available and they have been supplied together with the manuscript.
In these files it is clear that the Figures are very readable indeed.
This high resolution format has not carried over well into MS Word.
In the discussion, some sections (“literature review”) are in my opinion more suitable for the introduction section - but could stay in the discussion section with comparison or linkage to the study results (please add where relevant).
Response.
In line with this reviewer’s remarks we have greatly reduced the Discussion section.
Lines 533 – 657 in the tracked changes MS have been deleted as can readily be observed.
Moreover we have been careful to relate it to the main subject as much as possible.
I wonder if statistics were performed by a professional statistician or statistical department.
Response.
The statistics were performed by an academic statistician with many publications in high impact factor journals including journals of statistical methodology and public health [1-4].
References
- Reece AS, Hulse GK: Epidemiological association of cannabinoid- and drug- exposures and sociodemographic factors with limb reduction defects across USA 1989–2016: A geotemporospatial study. Spatial and Spatio-temporal Epidemiology 2022, 41:100480-100490.
- Reece A.S., Hulse G.K.: Geotemporospatial and Causal Inferential Epidemiological Overview and Survey of USA Cannabis, Cannabidiol and Cannabinoid Genotoxicity Expressed in Cancer Incidence 2003–2017: Part 1 – Continuous Bivariate Analysis. Archives of Public Health 2022, 80:99-133.
- Reece A.S., Hulse G.K.: Geotemporospatial and Causal Inferential Epidemiological Overview and Survey of USA Cannabis, Cannabidiol and Cannabinoid Genotoxicity Expressed in Cancer Incidence 2003–2017: Part 2 – Categorical Bivariate Analysis and Attributable Fractions. Archives of Public Health 2022, 80:100-135.
- Reece A.S., Hulse G.K.: Geotemporospatial and Causal Inferential Epidemiological Overview and Survey of USA Cannabis, Cannabidiol and Cannabinoid Genotoxicity Expressed in Cancer Incidence 2003–2017: Part 3 – Spatiotemporal, Multivariable and Causal Inferential Pathfinding and Exploratory Analyses of Prostate and Ovarian Cancers. Archives of Public Health 2022, 80:100-136.

Reviewer 2 Report
1. In general, there are many grammatical errors in the Abstract and the whole content. Many sentences need to have the comma to avoid too long to be understood easily. This manuscript must be polished by a native English speaker.
2. Please provide more information and citation about the E-values applied in this study.
3. Page 3, Lines 113-115: Funding, Conflicts of Interest, and Inform Consent should be moved to the end of the content.
4. Page 3, Line 120: The number “14,598” in eTable 1 should be the cancer cases rather than the “age-adjusted cancer incidence rates”. Please clarify this. Additionally, the ethnicity in eTable 1 has different statistical parameters for American, such as Caucasian American (mean (SD)) and African American (median [IQR]). Please declare the information.
5. Table 4: Please describe how to include different terms in the interactive panel model for different cancers?
6. The Discussion is too long and has to be brief and to the point.
7. Some risk factors related to different types of cancer, such as body mass index, hormone use/therapy, and regular exercise, are not adjusted in the model. This limitation may eliminate the validity of findings in the present study.
Author Response
Response to Reviewer 2.
Reviewer 2
- In general, there are many grammatical errors in the Abstract and the whole content. Many sentences need to have the comma to avoid too long to be understood easily. This manuscript must be polished by a native English speaker.
Response.
This has been done.
- Please provide more information and citation about the E-values applied in this study.
Response.
This has been done. Please see lines 141-146 where we now write:
“E-values, or expected values, allow a quantification of the extent to which an observed association might possibly be attributed to external or extraneous uncontrolled covariates [63, 64]. E-Values were calculated using the EValue R package [65]. The E-value has a confidence interval reflecting its upper and lower bound. Minimum E-Values in excess of 1.25 are said to indicate causality [66] and E-values exceeding nine are considered to be very high [67].”
- Page 3, Lines 113-115: Funding, Conflicts of Interest, and Inform Consent should be moved to the end of the content.
Response.
This has been done.
- Page 3, Line 120: The number “14,598” in eTable 1 should be the cancer cases rather than the “age-adjusted cancer incidence rates”. Please clarify this.
Response.
This statement is quite clear.
It is also consistent with eTable 1.
14,598 cancer incidence rates were downloaded.
This is NOT the case numbers.
Case numbers are provided in eTables 6-8 and Table 1.
Additionally, the ethnicity in eTable 1 has different statistical parameters for American, such as Caucasian American (mean (SD)) and African American (median [IQR]). Please declare the information.
Response.
Where covariates are normally distributed the mean and standard deviation are calculated.
Where covariates are not normally distributed the median and interquartile range is calculated.
Since most of the ethnic groups are not normally distributed it is appropriate to use the median and interquartile range.
The sole exception to this is the Caucasian American group which is normally distributed and fort this group the mean and standard deviation are presented.
- Table 4: Please describe how to include different terms in the interactive panel model for different cancers?
Response.
In all interactive models a three way interaction was introduced between tobacco, alcohol use disorder and Δ8THC exposure.
This is now explained with a comment at lines 140:
“In all interactive models a three way interaction was introduced between tobacco, alcohol use disorder, and Δ8THC exposure.”
- The Discussion is too long and has to be brief and to the point.
Response.
This comment is in contradistinction to the remarks of the first reviewer.
Moreover the cellular and molecular mechanisms likely to be involved are central to the causal argument.
Moreover they are not well understood even within the medical and research communities.
For these reasons we do feel that it is important to include at least some of the mechanistic discussion which has therefore been retained.
Nevertheless large sections have been removed from the Discussion consistent with this reviewer’s suggestion.
Lines 533 – 657 in the tracked changes MS have been deleted as can readily be observed.
- Some risk factors related to different types of cancer, such as body mass index, hormone use/therapy, and regular exercise, are not adjusted in the model. This limitation may eliminate the validity of findings in the present study.
Response.
We do not agree with this remark.
Whilst it is true that cancer specific risk factors such as those mentioned by this reviewer were not considered in this analysis the very high E-values reported exclude significant perturbation of the main results. The reviewer is respectfully referred to Tables 2 and 4 where most minimum E-values are infinite indicating that this would not affect the main findings. For these reasons we have now inserted a comment under the Limitations section at line 703 which reads:
“Whilst cancer-specific risk factors were not studied (such as hormonal exposure, body mass index and regular exercise) the very high E-values reported in this study (see Tables 2 and 4) indicate that inclusion of such covariates would not greatly perturb the main results reported herein.”

Round 2
Reviewer 2 Report
None.